# Maternal Seafood Consumption during Pregnancy and Cardiovascular Health of Children at 11 Years of Age

**DOI:** 10.3390/nu16070974

**Published:** 2024-03-27

**Authors:** Ariadna Pinar-Martí, Sílvia Fernández-Barrés, Iolanda Lázaro, Serena Fossati, Silvia Fochs, Núria Pey, Martine Vrijheid, Dora Romaguera, Aleix Sala-Vila, Jordi Julvez

**Affiliations:** 1Clinical and Epidemiological Neuroscience (NeuroÈpia), Institut d’Investigació Sanitària Pere Virgili (IISPV), 43204 Reus, Spain; ariadna.pinar@iispv.cat; 2ISGlobal-Instituto de Salud Global de Barcelona-Campus MAR, PRBB, 08003 Barcelona, Spain; silviabarres@gmail.com (S.F.-B.); serena.fossati@isglobal.org (S.F.); silvia.fochs@isglobal.org (S.F.); nuria.pey@isglobal.org (N.P.); martine.vrijheid@isglobal.org (M.V.); dora.romaguera@isglobal.org (D.R.); 3Departament de Medicina i Ciències de la Vida (MELIS), Universitat Pompeu Fabra (UPF), 08003 Barcelona, Spain; 4Cardiovascular Risk and Nutrition, Hospital del Mar Medical Research Institute (IMIM), 08003 Barcelona, Spain; ilazaro@researchmar.net (I.L.); asala3@imim.es (A.S.-V.); 5Centro de Investigación Biomédica en Red de Fisiopatología de la Obesidad y Nutrición (CIBEROBN), Instituto de Salud Carlos III, 28029 Madrid, Spain; 6Centro de Investigación Biomédica en Red de Epidemiología y Salud Pública (CIBERESP), Instituto de Salud Carlos III, 28029 Madrid, Spain; 7Health Research Institute of the Balearic Islands (IdISBa), 07120 Palma de Mallorca, Spain; 8Departament de Bioquímica i Biotecnologia, Unitat de Nutrició Humana, Universitat Rovira i Virgili, 43201 Reus, Spain

**Keywords:** cardiovascular health, omega-3, maternal nutrition, pregnancy, children health, fish intake during pregnancy, mediterranean diet

## Abstract

Nutrition is critical during pregnancy for the healthy growth of the developing infant, who is fully dependent on maternal dietary omega-3 supply for development. Fatty fish, a main dietary source of omega-3, is associated with decreased cardiovascular risk in adults. We conducted a longitudinal study based on a mother–offspring cohort as part of the project Infancia y Medio Ambiente (INMA) in order to assess whether fish intake during pregnancy relates to cardiovascular health in children. A total of 657 women were included and followed throughout pregnancy until birth, and their children were enrolled at birth and followed up until age 11–12. A semi-quantitative food frequency questionnaire was used to assess the daily intake of foods during the 1st and 3rd trimesters of pregnancy. Cardiovascular assessments included arterial stiffness (assessed by carotid–femoral pulse wave velocity [PWV]) and retinal microcirculation (photographic assessment of central retinal arteriolar and venular equivalent [CRAE and CRVE]). The association between maternal fish consumption and cardiovascular outcomes of offspring at 11 years of age was evaluated using multivariable linear regression models. There were no statistically significant differences in any cardiovascular endpoint in children whose mothers had a higher fish consumption during pregnancy compared to those with a lower fish consumption. We found a slightly higher PWV (β = 0.1, 95% CI = 0.0; 0.2, *p* for trend = 0.047) in children whose mothers had a higher consumption of canned tuna during the 1st trimester of pregnancy. Fish intake during pregnancy was found to be unrelated to the offspring’s cardiovascular health at 11 years of age. The beneficial cardiovascular effects of fish consumption during pregnancy on the offspring are still inconclusive.

## 1. Introduction

Cardiovascular diseases (CVDs) are the leading cause of mortality worldwide, causing about 17.9 million deaths per year [1,2]. However, most CVDs can be prevented by addressing behavioral risk factors, such as an unhealthy diet. Indeed, recent studies have shown that high adherence to healthy diets (i.e., Mediterranean diet) can help minimize the risk of cardiovascular disease [3,4,5].

The Mediterranean diet is well known for its abundance of fish. Sustained consumption of fatty fish, the main source of *n*-3 eicosapentaenoic acid (EPA) and *n*-3 docosahexaenoic acid (DHA), reduces cardiovascular risk in adults [6,7,8]. In fact, the prevalence of CVD is lower in populations with high fish consumption [9,10,11,12,13]. This could largely be due to the anti-inflammatory, antiarrhythmic, and antihypertensive effects of EPA and DHA [8,12]. Fish can also contain additional nutrients that provide health benefits and cardioprotection, such as vitamin D, calcium, selenium, or proteins [14,15,16].

While the cardioprotective benefits of a fish-rich diet have been extensively studied, whether maternal fish intake during pregnancy affects later offspring vascular health remains overlooked [17]. However, it is well understood that nutrition is critical throughout pregnancy for the proper and healthy growth of the developing infant [18], and the fetus is fully dependent on maternal dietary *n*-3 supply for development [19]. Clinical research on the effects of fish intake during pregnancy on the offspring’s cardiovascular health is limited to a single study conducted in 234 mother–offspring pairs from the UK, wherein the child’s aortic pulse wave velocity (PWV) at the age of 9 years was inversely related to self-reported maternal fatty fish consumption in early or late pregnancy [20]. Further, one randomized clinical trial found that some inflammatory and vascular homeostasis biomarkers changed during pregnancy but were unaffected by an increase in farmed salmon consumption [21]. Given the significant findings of these studies and because research in this field is very limited, this line of research warrants further exploration.

Higher carotid–femoral PWV directly reflects increased aortic stiffness, a strong predictor of future cardiovascular events [22,23]. Similarly, biomarkers of retinal microcirculation, such as narrowing of central retinal arteriolar equivalent (CRAE) and widening of central retinal venular equivalent (CRVE), are often associated with cardiovascular risk and a higher incidence of CVD [24,25,26].

We conducted a prospective study in order to explore whether maternal fish intake during the 1st and 3rd trimesters of pregnancy was associated with the cardiovascular health of the 11-year-olds, including arterial stiffness (as assessed by PWV) and retinal microcirculation (photographic assessment of CRAE and CRVE). Further, we also searched for specific associations with different types of consumed fish (lean fish; large fatty fish; small fatty fish; shellfish; and canned tuna). Our main hypothesis was that eating more fish during pregnancy would result in lower arterial stiffness (lower PWV), narrower CRAE, and wider CRVE in offspring at 11 years old, resulting in better cardiovascular health.

## 2. Materials and Methods

### 2.1. Study Design and Participants

This is a longitudinal study based on a mother–offspring cohort established in the city of Sabadell as part of the project Infancia y Medio Ambiente (INMA) [27,28]. The birth cohort INMA-Sabadell included women who attended the CAP II Sant Félix of Sabadell between July 2004 and July 2006 between the 4th and the 6th week of pregnancy. The inclusion criteria were 16 years of age or older, single pregnancy, not having followed an assisted reproduction program, and free of chronic diseases at the time of conception. A total of 657 women were included, who were followed every trimester of the pregnancy until the moment of birth, and their children were enrolled at birth and followed up until age 11–12. The present analysis was limited to children who participated in the 11-year follow-up, who had at least one of the measured cardiovascular outcomes, and whose mothers had dietary data during their 1st or 3rd trimester of pregnancy (N = 434).

### 2.2. Sociodemographic, Clinical, and Lifestyle Data

Questionnaires collecting information on maternal and child characteristics were administered by trained interviewers twice during pregnancy (12 and 32 weeks) and at the child’s age of 11 years. We obtained data from maternal education and social class, smoking habits during pregnancy, parity, maternal pre-pregnancy body mass index (BMI), mother’s age during pregnancy, and parental cardiovascular history (neither parent has a diagnosis/one parent has at least one diagnosis/both parents have at least one diagnosis) based on questionnaire data from parents on previous cardiovascular events: heart attack, angina, hemorrhage or stroke, arthrosclerosis in legs, high cholesterol, high blood sugar, and high blood pressure [29].

A clinical examination of the children was also performed to collect anthropometric measurements. Height and weight were measured by standard methods. Systolic and diastolic blood pressure was also assessed using an automatic digital monitor (OMRON 705IT, Omron, Kyoto, Japan) with brachial cuff attached. After 5 min of rest, three consecutive measurements were taken with one-minute time intervals between them. Subsequently, mean blood pressure was calculated.

A nurse obtained blood samples after an overnight fast. The samples were stored at 4 °C for 1 h and were then centrifuged at 2500× *g* for 15 min into serum. Low-density lipoprotein cholesterol (LDL-C) was determined by a direct method (particle elimination, cholesterol esterase, and colorimetry) [30].

### 2.3. Seafood Consumption

A semi-quantitative food frequency questionnaire (FFQ) of 101 food items was used to assess the usual daily intake of foods and nutrients during the 1st and 3rd trimesters of pregnancy. The FFQ was a modified version of a previous FFQ based on the Harvard questionnaire [31], adapted and validated among the pregnant women of the Sabadell cohort [32]. Women reported their usual intake of foods using reference portions and nine frequency categories ranging from never/less than once a month to more than six times per day. The questionnaire included 11 seafood items. The response to each seafood item was converted to average weekly (w) intakes in grams (g) and then summed to compute the total and seafood subtypes (in g/w). Seafood was classified a priori as follows: (i) large fatty fish, including one item from the questionnaire (“baked or steamed large fatty fish such as tuna, swordfish, albacore”); (ii) small fatty fish, including two items from the questionnaire (“baked or steamed small fatty fish such as mackerel, sardines, anchovies, salmon”; “tinned sardines/mackerel”); (iii) canned tuna, one item; (iv) lean fish, including two items (“varied fried fish”; “baked or steamed lean fish such as hake, sole, or bream”); (v) shellfish, including three items (“shrimp, prawns, lobster or crab”; “clams, mussels, oysters”; “squid, octopus, cuttlefish”); (vi) processed fish (i.e., surimi), one item; (vii) smoked/salted fish, including one item (“salted or smoked fish: anchovies, cod, salmon”); and (viii) overall (total) seafood intakes, calculated as the sum of consumption of all previous items. Processed fish and smoked/salted fish were excluded from the main analyses as subtypes due to their low intake frequency but included in the overall seafood intakes. Intakes were adjusted for energy intake using the residual method and analyzed in tertile categories of weekly grams for total seafood.

To collect the diet of the children at 11 years of age, a short questionnaire was answered by the mothers [33]. The frequency of fish consumption, assessed as servings per week, included 7 seafood items: (i) large fatty fish; (ii) small fatty fish, (iii) lean fish; (iv) canned sardine or mackerel; (v) canned tuna; (vi) shellfish; and (vii) processed fish (i.e., surimi).

### 2.4. Cardiovascular Endpoints

Cardiovascular measurements were conducted by trained nurses at the school during the clinical examination. The PWV was measured using VICORDER^®^ (https://www.medallianceinternational.com/product/vicorder-vascular-testing-system/ accessed on 24 March 2024), an arterial stiffness testing system with a neck cuff (20 mm width) and a femoral cuff (100 mm width) attached in combination with the VICORDER^®^ vascular diagnostic program package. PWV is an established cardiovascular risk marker, and it is considered the gold standard measurement of arterial stiffness [22,34]. Arterial stiffness measurement as PWV in children by VICORDER^®^ has been validated in children against the non-invasive gold standard (i.e., applanation tonometry) as the American Heart Association (AHA) recommends [35]. Children were asked to lay in the supine position with support to raise their heads and shoulders 30° above their heart level. The thigh cuff was placed on the upper right thigh as high as possible, and the neck cuff was placed after palpating the pulse of the right common carotid artery in the center between the base of the neck and chin. The 80% method was used (80% of the measured direct distance between the carotid and femoral recording sites) to ensure the smallest possible measurement error since, in the measurement of PWV, the major source of inaccuracy lies in the determination of the travel distance of the pulse wave. PWV measurements were recorded in triplicate measurements taken in a row. The mean values of the three measurements were used for further analyses. Higher PWV reflects increased arterial stiffness. Values are given in meters per second (m/s) [29].

As for the retinal vascular caliber assessment, retinal images were photographed using a Canon CR2-plus Non-Mydriatic Retinal camera, which provides a non-invasive, in vivo method for characterizing the human microvasculature (retinal vessels are 60–300 μm in diameter). The child was asked to sit behind the camera with their chin on the chin rest and their forehead pressed to the overhead bar and was asked to look straight into the camera lens while the operator took the retina images. Photos were taken from both the left and right eyes and saved in high resolution for later analysis to calculate CRVE and CRAE. Average CRVE and CRAE were calculated based on an average of the six widest arterioles and venules running through a zone between 0.5 and 1 disk diameter from the optic disk margin. CRAE and CRVE measures were averaged between the right and left eyes. Narrower CRAE and wider CRVE correspond to a higher cardiovascular risk. Values are given in micrometers (μm).

### 2.5. Statistical Analysis

Descriptive statistics were used to display the baseline characteristics of the study population according to maternal fish intake (in tertiles) during the 1st and 3rd trimesters of pregnancy. Categorical variables are expressed as percentages and continuous variables are expressed as means (standard deviation [SD]). The association between total maternal fish consumption and intake of different subtypes of seafood during the 1st and 3rd trimester of pregnancy and cardiovascular outcomes of offspring at 11 years of age were evaluated using multivariable linear regression models. Exposures were all seafood (main exposure) and isolated types of seafood, including large fatty fish, small fatty fish, canned tuna, lean fish, and shellfish (secondary exposures). Exposures were evaluated as ordinal variables (tertiles), and cardiovascular endpoints were evaluated as continuous variables. For all analyses, two models were built between the exposure and the outcomes: first, a minimally adjusted model with child age (years), child sex (male/female), maternal education (primary or less/secondary/university studies), maternal social class (low/medium/high), and total energy intake (kcal/d); secondly, a fully adjusted model with all confounders included, which was further adjusted for maternal age (years), parental cardiovascular history (neither parent has diagnosis/one parent has at least one diagnosis/both parents have at least one diagnosis), parity (0/1/2 or more), maternal smoking during pregnancy (yes/no), and maternal pre-pregnancy BMI (kg/m^2^). The main confounders were selected according to previous scientific knowledge and then using a Directed Acyclic Graph (DAG) model (Appendix A).

Several sensitivity analyses on the association of 1st and 3rd trimesters total maternal fish intake and cardiovascular outcomes of offspring at 11 years of age were conducted additionally adjusting for (i) child height; (ii) child age-and-sex-specific z-score BMI; (iii) child blood pressure at 11 years of age; (iv) child low-density lipoprotein (LDL)-cholesterol at 9 years of age; and (v) child fish intake at 11 years of age (servings/week).

The nominal statistical significance was set at the *p* < 0.05 level (two-sided) for the primary outcomes. Statistical analyses were conducted using the STATA 15 statistical software package (Stata Statistical Software: Release 15, StataCorp LLC: College Station, TX, USA, 2017).

## 3. Results

Table 1 and Table 2 display baseline characteristics of the study population by tertiles of maternal fish consumption. The mean maternal age was 32 ± 4 years; the majority of the mothers did not smoke during pregnancy (88%) and had a normal pre-pregnancy BMI (23.8 ± 4.6 kg/m^2^). Approximately 44% of children had one parent with at least one cardiovascular history event (i.e., heart attack, angina, hemorrhage or stroke, arthrosclerosis in legs, high cholesterol, high blood sugar, and high blood pressure). Maternal energy intake was significantly higher in participants reporting high consumption of fish during the 1st and 3rd trimesters of pregnancy compared to those reporting lower consumption. At the time of cardiovascular measurements, children were, on average, 11 years of age, and there were no differences in the distribution of genders among tertiles of maternal fish consumption. Child fish consumption was also significantly higher in participants whose mothers reported higher consumption of fish during the 1st and 3rd trimesters of pregnancy.

The median maternal total seafood consumption was 451.9 g/week (IQR = 345.5) during the 1st trimester of pregnancy and 433.8 g/week (IQR = 315.4) during the 3rd trimester of pregnancy (Appendix A).

The average CRAE and CRVE in children 11 years of age were 181.1 ± 12.8 μm and 252.2 ± 17.3 μm, respectively, and PWV was 4.38 ± 0.46 m/s (Appendix A).

Minimally and fully adjusted regression models for associations of maternal total fish and subtype fish consumption for the 1st and 3rd trimesters of pregnancy and offspring’s cardiovascular endpoints at 11 years of age are shown in Table 3, Table 4, Table 5 and Table 6. There were no significant differences in any cardiovascular endpoint in children whose mothers had a higher fish consumption during pregnancy, either in the 1st or 3rd trimester, compared to those with a lower fish consumption. We found a slightly higher PWV (β = 0.1, 95% CI = 0.0; 0.2, *p* for trend = 0.047 for fully adjusted model) in children whose mothers had a higher consumption of canned tuna (3rd tertile; median, 150.1 g/w) compared to those with lower consumption (1st tertile; median, 23.4 g/w) on the 1st trimester of pregnancy (Table 5). Sensitivity analyses did not show any significant differences between tertiles of total maternal fish intake during 1st and 3rd trimesters of pregnancy and cardiovascular outcomes in offspring at 11 years of age (Appendix A).

## 4. Discussion

In this birth cohort study conducted within the framework of the INMA project [27], no significant associations in arterial stiffness and retinal vascular caliber were found in 11-year-old adolescents related to maternal fish consumption during the 1st and 3rd trimesters of pregnancy. There was no evidence of association for minimally adjusted models and this association did not change when further adjusting for maternal and paternal cardiovascular covariates. Similarly, a sensitivity analysis further adjusting for the child’s cardiovascular covariates also did not show any significant association with arterial stiffness and retinal microcirculation.

Data regarding the study of the beneficial effects of maternal fish consumption on cardiovascular health in adolescents are very limited. To the best of our knowledge, there is only one other clinical research that studied the effects of fish intake during pregnancy on the offspring’s cardiovascular health at a pre-adolescent age [20]. In contrast to our results, they found that higher consumption of oily fish in late pregnancy was associated with lower aortic stiffness in offspring at 9 years of age, although no association between the child’s current oily fish consumption and vascular stiffness was found.

There are, nonetheless, some possible explanations for our findings. First, because this is a young and healthy group, it is probable that the minor variances between individuals are not enough to observe significant differences. It is conceivable that alterations will probably begin to appear later in life. Most studies suggesting a reduction in PWV tend to occur in adults with pre-existing pathologies related to cardiovascular health (i.e., obesity, metabolic syndrome, diabetes, high blood pressure, cholesterol, or triglycerides) and are based on chronic fish-oil supplementation with long-chain *n*-3 polyunsaturated fatty acids (PUFAs) such as EPA and DHA [36]. In fact, when we talk about daily *n*-3 supplementation in healthy adults, there appears to be a significant decrease in central arterial stiffness (PWV) in older adults but not in healthy young people [37]. Thus, it is important to note that “arterial aging” has a substantial role in the progression of CVD with age [38].

Second, even though this is a high fish consumption cohort, there is a possibility that mercury exposure may have reduced the cardiovascular benefits of fish intake. Whether eating fish contaminated with mercury increases the risk of cardiovascular disease remains controversial [39,40,41], but the balance between mercury toxicity and the positive effects of fish consumption warrants more exploration. Further, precisely because this is a high fish consumption cohort, it is likely that there is a sufficiently high fish intake in all tertiles of fish consumption that there are no differences across groups. According to the AHA, the greatest difference occurs between 0 and 2 servings, with no benefit reported above [42]. In this scenario, the average fish consumption was 482.5g (4–5 servings), indicating that these women are probably above the protective threshold and that even those in the lowest tertile eat enough fish to obtain its benefits, so there are no significant differences between groups.

On the other hand, short-term effects of fish oil supplementation on cardiovascular health have been shown in infants in terms of blood pressure and heart rate [43,44]. Nonetheless, previous research into the effect of *n*-3 intake during pregnancy on later cardiovascular health is relatively sparse and has yielded inconclusive results, which is consistent with our findings. A longitudinal study found no association between maternal intake of marine *n*-3 PUFA during the second trimester of pregnancy and factors associated with cardiometabolic risk (i.e., blood pressure and heart rate) in the 20-year-old offspring [45]. Similarly, another longitudinal study found no effect of maternal fish oil supplementation during the last trimester of pregnancy on blood pressure, heart rate, and heart rate variability in the 19-year-old offspring [46]. A randomized controlled trial carried out in Denmark also found no associations between fish oil supplementation during the first four months of lactation and cardiovascular risk markers in children at 2.5 years of age, including PWV [47]. Thus, evidence for a possible long-chain *n*-3 PUFA programming effect on later cardiovascular health in humans is limited and inconclusive.

In summary, there are probably several important factors that need to be considered: (1) both age and baseline health conditions influence arterial status; (2) probably, a weekly fish consumption of two servings or more will provide significant benefits over no consumption but will provide little further benefits for greater servings; (3) more research is needed to determine the optimal balance between mercury toxicity and the benefits of fish consumption; and (4) evidence regarding the effect of *n*-3 intake during pregnancy on later cardiovascular health remains inconclusive.

This study has several strengths, such as the exploration of not only total fish intake but also several seafood subtypes. Further, the INMA study’s robust follow-up allowed us to measure cardiovascular endpoints in a pediatric population that is otherwise understudied. Nevertheless, this study faced some limitations. First, because of its observational nature, causality cannot be inferred. Second, despite being a validated tool to assess dietary intakes, the use of an FFQ is susceptible to measurement errors, which may lead to an attenuation of the effect estimate. For the same reason, fatty acids could have been quantified from cord blood samples to assess maternal blood *n*-3 unsaturated fatty acids more accurately than an FFQ. Although it was quantified in a small subsample, there were not enough cases to be included in the present study. Additionally, we cannot rule out the possibility of some residual confounding due to uncaptured environmental factors. However, we carefully considered a variety of potential confounders and performed sensitivity analyses to address this potential limitation. Third, a power analysis should have been performed a priori since there is a chance that this study lacks sufficient power to detect any associations, which might have been detected with a larger sample. Lastly, in the measurement of PWV, the major source of inaccuracy lies in the determination of the travel distance of the pulse wave, so there is always a possibility of measurement error. However, to ensure the smallest possible measurement error, the 80% method was used (80% of the measured direct distance between the carotid and femoral recording sites). Additionally, PWV was measured three times, and the average was then used for analyses. Further, it is important to consider that children’s blood vessels are still undergoing a growth phase, so there may be some questions as to whether this technology can accurately assess the cardiovascular risk of an 11-year-old child. For a better and more complete determination of cardiovascular health at 11 years of age, future studies should include echocardiography to assess heart health by observing heartbeat and heart valve motion, electrocardiograms to assess problems such as arrhythmias and heart dilation and exercise stress tests to assess the effects of exercise on the heart and to check for cardiovascular risk. Of note, follow-ups are important in epidemiological studies. Thus, future cardiovascular risk tracking studies should include regular health checks, exercise tests, blood tests, and lifestyle monitoring from childhood through adolescence.

## 5. Conclusions

Overall, we found that fish intake during the 1st and 3rd trimesters of pregnancy did not have any important associations with the offspring’s cardiovascular health at 11 years of age. The beneficial cardiovascular effects of fish consumption during pregnancy on the offspring are still inconclusive. This study provides useful insights and a foundation for future clinical and epidemiological research on the influence of maternal fish intake on children’s cardiovascular health.

## Figures and Tables

**Table 1 nutrients-16-00974-t001:** Characteristics of participants by tertiles of maternal fish intake (1st trimester of pregnancy) in INMA Sabadell cohort/study.

	All Participants (N = 432)	% Missings	T_1_ (N = 147, 33.9%)	T_2_ (N = 142, 32.7%)	T_3_ (N = 143, 32.9%)	*p*
Child Sex (%)		0				0.599
Female	210 (49)		71 (48)	65 (46)	74 (52)	
Male	224 (51)		76 (52)	77 (54)	69 (48)	
Child age at measurements, years	11.1 (0.5)	0.5	11.1 (0.5)	11.1 (0.6)	11.2 (0.5)	0.688
Child height at age of measurements, cm	146.8 (7.9)	0.5	146.8 (7.6)	147.2 (8.2)	146.5 (7.9)	0.761
Child zBMI at age of measurements, years	0.7 (1.2)	0.5	0.7 (1.2)	0.6 (1.2)	0.7 (1.3)	0.704
Child fish intake at age of measurements, sv/w	1.6 (1.3)	1.8	1.4 (1.2)	1.4 (1.1)	2.0 (1.5)	<0.001
Parental Cardiovascular History (%)		1.4				0.441
None	199 (47)		68 (47)	62 (45)	69 (48)	
1 parent has ≥1 diagnosis	188 (44)		65 (45)	66 (47)	56 (39)	
Both parents have ≥1 diagnosis	40 (9)		11 (8)	11 (8)	18 (13)	
Maternal age, years	31.8 (4.2)	0	31.0 (4.3)	32.2 (4.2)	32.4 (3.8)	0.009
Maternal Pre-pregnancy BMI, kg/m^2^	23.8 (4.6)	0	24.0 (4.9)	23.6 (4.4)	23.8 (4.4)	0.743
Maternal energy intake, kcal/d	2047.8 (485.7)	0	1917.2 (483.8)	2017.9 (422.0)	2211.7 (502.9)	<0.001
Maternal Smoking in pregnancy (%)		1.6				0.218
No	375 (88)		122 (84)	127 (91)	126 (89)	
Yes	52 (12)		23 (16)	13 (9)	16 (11)	
Maternal Education level (%)		0.5				0.205
Primary or less	102 (24)		38 (26)	32 (23)	32 (23)	
Secondary	183 (43)		69 (47)	52 (37)	62 (44)	
University	145 (34)		40 (27)	57 (40)	48 (34)	
Maternal Social Class (%)		0				0.239
Low	104 (24)		32 (22)	36 (25)	36 (25)	
Medium	145 (34)		42 (29)	54 (38)	49 (34)	
High	183 (42)		73 (50)	52 (38)	49 (34)	
Parity (%)		0.5				0.365
0	252 (59)		92 (63)	76 (54)	84 (59)	
1	155 (36)		48 (33)	58 (41)	49 (34)	
≥2	23 (5)		5 (3)	8 (6)	10 (7)	

zBMI: body mass index z-scores; sv/w: servings/week. Values shown are mean (SD) unless otherwise specified. Sample population was limited to the ones with maternal fish intake during the 1st trimester and at least one cardiovascular outcome of children at 11 years old.

**Table 2 nutrients-16-00974-t002:** Characteristics of participants by tertiles of maternal fish intake (3rd trimester of pregnancy) in INMA Sabadell cohort/study.

	All Participants (N = 430)	% Missings	T_1_ (N = 146, 33.6%)	T_2_ (N = 150, 34.6%)	T_3_ (N = 134, 30.9%)	*p*
Child Sex (%)		0				0.332
Female	207 (48)		63 (43)	76 (51)	68 (51)	
Male	223 (52)		83 (57)	74 (49)	66 (49)	
Child age at measurements, years	11.2 (0.5)	0.5	11.1 (0.5)	11.2 (0.5)	11.2 (0.5)	0.486
Child height at age of measurements, cm	146.8 (7.9)	0.5	147.3 (8.0)	146.2 (7.6)	146.9 (8.1)	0.464
Child zBMI at age of measurements, years	0.7 (1.2)	0.5	0.8 (1.2)	0.7 (1.2)	0.6 (1.3)	0.534
Child fish intake at age of measurements, sv/w	1.6 (1.3)	1.9	1.4 (1.0)	1.6 (1.4)	1.9 (1.4)	0.004
Paternal Cardiovascular History (%)		1.4				0.35
None	198 (47)		58 (41)	72 (49)	68 (51)	
1 parent has ≥1 diagnosis	186 (44)		72 (50)	63 (43)	51 (38)	
Both parents have ≥1 diagnosis	40 (9)		13 (9)	13 (9)	14 (11)	
Maternal Smoking in pregnancy (%)		0.5				0.559
No	376 (88)		124 (86)	133 (89)	119 (89)	
Yes	52 (12)		21 (14)	17 (11)	14 (11)	
Maternal Education level (%)		0.7				0.497
Primary or less	100 (23)		33 (23)	32 (22)	35 (26)	
Secondary	184 (43)		57 (39)	68 (46)	59 (44)	
University	143 (33)		56 (38)	48 (32)	39 (29)	
Maternal Social Class (%)		0				0.613
Low	103 (24)		38 (26)	39 (26)	26 (19)	
Medium	145 (34)		45 (31)	51 (34)	49 (37)	
High	182 (42)		63 (43)	60 (40)	59 (44)	
Maternal energy intake, kcal/d	2081.0 (467.4)	0.5	1909.9 (441.0)	2103.7 (435.3)	2242.1 (470.5)	<0.001
Maternal Pre-pregnancy BMI, kg/m^2^	23.8 (4.6)	0	24.0 (4.5)	23.8 (4.8)	23.6 (4.3)	0.781
Maternal age, years	31.8 (4.2)	0.2	31.4 (4.6)	32.0 (4.1)	32.1 (3.8)	0.241
Parity (%)		0.5				0.4
0	249 (58)		88 (61)	90 (60)	71 (53)	
1	155 (36)		50 (35)	49 (33)	56 (42)	
≥2	24 (6)		6 (4)	11 (7)	7 (5)	

zBMI: body mass index z-scores; sv/w: servings/week. Values shown are mean (SD) unless otherwise specified. Sample population was limited to the ones with maternal fish intake during the 3rd trimester and at least one cardiovascular outcome of children/offspring at 11 years old.

**Table 3 nutrients-16-00974-t003:** Association between tertiles of maternal fish intake (1st trimester of pregnancy) and cardiovascular outcomes in INMA Sabadell children/offspring at 11 years of age.

		T_1_ (N = 147, 33.9%)	T_2_ (N = 142, 32.7%)	T_3_ (N = 143, 32.9%)	
Outcome	N	β (95% CI)	β (95% CI)	β (95% CI)	*p* for Trend
CRAE					
Model 1	397	Ref	−1.3 (−4.5, 1.8)	−1.7 (−4.9, 1.4)	0.280
Model 2	386	Ref	2.1 (−5.3, 1.1)	−1.9 (−5.2, 1.3)	0.245
CRVE					
Model 1	397	Ref	−1.5 (−5.8, 2.7)	0.2 (−4.2, 4.5)	0.952
Model 2	386	Ref	−1.6 (−5.9, 2.8)	1.0 (−3.4, 5.5)	0.642
PWV					
Model 1	411	Ref	0.0 (−0.1, 0.1)	0.0 (−0.1, 0.1)	0.778
Model 2	400	Ref	0.0 (−0.1, 0.1)	0.0 (−0.1, 0.1)	0.898

CI: confidence interval; CRAE: central retinal artery equivalent; CRVE: central retinal vein equivalent; PWV: pulse wave velocity. Data obtained by linear regression analysis. Model 1: adjusted for child age, child sex, maternal education, maternal social class, and total energy intake; Model 2: Model 1 further adjusted for maternal age, paternal cardiovascular history, parity, maternal smoking during pregnancy, and maternal pre-pregnancy body mass index.

**Table 4 nutrients-16-00974-t004:** Association between tertiles of maternal fish intake (3rd trimester of pregnancy) and cardiovascular outcomes in INMA Sabadell children/offspring at 11 years of age.

		T_1_ (N = 146, 33.6%)	T_2_ (N = 150, 34.6%)	T_3_ (N = 134, 30.9%)	
Outcome	N	β (95% CI)	β (95% CI)	β (95% CI)	*p* for Trend
CRAE					
Model 1	394	Ref	1.1 (−2.0, 4.2)	1.0 (−2.2, 4.3)	0.534
Model 2	386	Ref	0.6 (−2.6, 3.7)	0.9 (−2.5, 4.2)	0.605
CRVE					
Model 1	394	Ref	0.3 (−3.9, 4.5)	0.0 (−4.4, 4.5)	0.242
Model 2	386	Ref	0.1 (−4.2, 4.4)	0.4 (−4.1, 5.0)	0.849
PWV					
Model 1	408	Ref	0.0 (−0.2, 0.1)	0.0 (−0.1, 0.1)	0.913
Model 2	400	Ref	0.0 (−0.1, 0.1)	0.0 (−0.1, 0.1)	0.800

CI: confidence interval; CRAE: central retinal artery equivalent; CRVE: central retinal vein equivalent; PWV: pulse wave velocity. Data obtained by linear regression analysis. Model 1: adjusted for child age, child sex, maternal education, maternal social class, and total energy intake; Model 2: Model 1 further adjusted for maternal age, paternal cardiovascular history, parity, maternal smoking during pregnancy, and maternal pre-pregnancy body mass index.

**Table 5 nutrients-16-00974-t005:** Association between tertiles of types of maternal fish intake (1st trimester of pregnancy) and cardiovascular outcomes in INMA Sabadell children/offspring at 11 years of age (N = 400).

	CRAE	CRVE	PWV
	β (95% CI)	β (95% CI)	β (95% CI)
Large fatty fish			
Tertiles: 1 (median, 0 g/w)	Ref	Ref	Ref
2 (median, 47.1 g/w)	0.3 (−3.1, 3.7)	−0.3 (−5.0, 4.3)	0.0 (−0.1, 0.1)
3 (median, 100.5 g/w)	2.0 (−1.3, 5.3)	0.4 (−4.0, 4.9)	−0.1 (−0.2, 0.0)
*p* for trend	0.255	0.888	0.143
Small fatty fish			
Tertiles: 1 (median, 0 g/w)	Ref	Ref	Ref
2 (median, 4.7 g/w)	0.7 (−2.5, 3.9)	0.8 (−3.5, 5.1)	0.0 (−0.1, 0.1)
3 (median, 100.5 g/w)	1.1 (−2.0, 4.2)	2.9 (−1.3, 7.0)	0.0 (−0.1, 0.1)
*p* for trend	0.487	0.178	0.828
Canned Tuna			
Tertiles: 1 (median, 23.4 g/w)	Ref	Ref	Ref
2 (median, 50.5 g/w)	−1.4 (−4.7, 1.8)	−0.7 (−5.1, 3.7)	0.1 (−0.0, 0.2)
3 (median, 150.1 g/w)	−1.4 (−4.7, 1.8)	−2.1 (−6.5, 2.3)	0.1 (0.0, 0.2)
*p* for trend	0.392	0.350	0.047
Lean fish			
Tertiles: 1 (median, 94.2 g/w)	Ref	Ref	Ref
2 (median, 201.0 g/w)	0.3 (−2.7, 3.4)	4.6 (0.5, 8.7)	0.0 (−0.1, 0.1)
3 (median, 402.0 g/w)	0.7 (−2.7, 4.2)	3.0 (−1.6, 7.6)	0.0 (−0.2, 0.1)
*p* for trend	0.678	0.136	0.458
Shellfish			
Tertiles: 1 (median, 23.6 g/w)	Ref	Ref	Ref
2 (median, 72.4 g/w)	−1.0 (−4.2, 2.1)	−0.6 (−4.9, 3.7)	0.0 (−0.1, 0.1)
3 (median, 104.5 g/w)	−1.8 (−5.1, 1.4)	−2.0 (−6.4, 2.4)	0.0 (−0.1, 0.2)
*p* for trend	0.273	0.366	0.464

CI: confidence interval; CRAE: central retinal artery equivalent; CRVE: central retinal vein equivalent; PWV: pulse wave velocity. Data obtained by linear regression analysis. Model adjusted for child age, child sex, maternal education, maternal social class, total energy intake, maternal age, paternal cardiovascular history, parity, maternal smoking during pregnancy and maternal pre-pregnancy body mass index.

**Table 6 nutrients-16-00974-t006:** Association between tertiles of types of maternal fish intake (3rd trimester of pregnancy) and cardiovascular outcomes in INMA Sabadell children/offspring at 11 years of age (N = 400).

	CRAE	CRVE	PWV
	β (95% CI)	β (95% CI)	β (95% CI)
Large fatty fish			
Tertiles: 1 (median, 0 g/w)	Ref	Ref	Ref
2 (median, 47.1 g/w)	−1.1 (−4.6, 2.5)	−0.1 (−5.0, 4.7)	−0.1 (−0.2, 0.1)
3 (median, 100.5 g/w)	2.0 (−1.2, 5.2)	2.6 (−1.8, 7.0)	0.0 (−0.1, 0.1)
*p* for trend	0.313	0.291	0.651
Small fatty fish			
Tertiles: 1 (median, 0 g/w)	Ref	Ref	Ref
2 (median, 47.1 g/w)	1.3 (−2.0, 4.5)	1.5 (−3.0, 5.9)	0.1 (0.0, 0.2)
3 (median, 100.5 g/w)	0.9 (−2.2, 4.0)	0.7 (−3.5, 4.9)	0.1 (0.0, 0.2)
*p* for trend	0.555	0.721	0.157
Canned Tuna			
Tertiles: 1 (median, 23.4 g/w)	Ref	Ref	Ref
2 (median, 50.5 g/w)	0.1 (−3.0, 3.3)	−1.2 (−5.5, 3.0)	0.0 (−0.1, 0.1)
3 (150.1 g/w)	0.6 (−2.8, 4.0)	−2.7 (−7.3, 1.9)	0.0 (−0.1, 0.1)
*p* for trend	0.732	0.234	0.702
Lean fish			
Tertiles: 1 (median, 94.2 g/w)	Ref	Ref	Ref
2 (median, 201.0 g/w)	−0.4 (−3.3, 2.6)	0.4 (−3.5, 4.4)	0.0 (−0.1, 0.1)
3 (median, 402.0 g/w)	−0.9 (−4.5, 2.8)	−0.2 (−5.1, 4.8)	−0.1 (−0.2, 0.1)
*p* for trend	0.647	0.991	0.274
Shellfish			
Tertiles: 1 (median, 23.6 g/w)	Ref	Ref	Ref
2 (median, 72.4 g/w)	−1.0 (−4.0, 2.1)	1.4 (−2.7, 5.6)	0.0 (−0.1, 0.1)
3 (median, 101.1 g/w)	1.0 (−2.2, 4.3)	0.6 (−3.8, 5.0)	0.1 (−0.1, 0.2)
*p* for trend	0.591	0.741	0.355

CI: confidence interval; CRAE: central retinal artery equivalent; CRVE: central retinal vein equivalent; PWV: pulse wave velocity. Data obtained by linear regression analysis. Model adjusted for child age, child sex, maternal education, maternal social class, total energy intake, maternal age, paternal cardiovascular history, parity, maternal smoking during pregnancy and maternal pre-pregnancy body mass index.

## Data Availability

Data described in the manuscript, code book, and analytic code will be made available upon request for checking and revising it after following internal rules and proceedings (DTA agreement) of INMA project.

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
