# Peer review of "Maternal Seafood Consumption during Pregnancy and Cardiovascular Health of Children at 11 Years of Age"

_nutrients, 2024, doi:10.3390/nu16070974_

Round 1
Reviewer 1 Report
Comments and Suggestions for Authors
The association between maternal gestational diet and future cardiovascular outcomes in the unborn offspring is a very interesting study. In this study, the authors evaluated fish intake during pregnancy and cardiovascular risk at age 11 years in healthy mothers. In many studies in this area have reported associations with fish oil and n-3 unsaturated fatty acids in cases of poor maternal health, and the difficulty of assessing cardiovascular risk at age 11 in analyses of healthy pregnant women with different genetic and other environmental factors is understandable. However, there are several fundamental problems with the study that may mislead readers of this paper. It should be added, however, that the background of the study is an important perspective on the problem of mercury contamination of the oceans, which has become an issue in recent years, including the problem of excessive fish consumption because it is good for the body. It is necessary to reconsider the following points and continue this research as a follow-up study through adulthood.
Comment 1.
The accuracy of the Food Frequency Questionnaire (FFQ) is uncertain and n-3 unsaturated fatty acids should be quantified from samples of gestational blood and cord blood to assess maternal blood n-3 unsaturated fatty acids.
Comment 2.
Cardiovascular outcomes at age 11 are naturally influenced by environmental and lifestyle factors from birth. Diet during pregnancy is important, but other factors (exercise, dietary changes, family genetic factors, etc.) must also be considered and evaluated. In addition, many questions remain as to whether the progression of future atherosclerosis can be assessed at age 11.
Comment 3.
Arterial stiffness testing with the VICORDER® is typically used to assess cardiovascular risk in adults. In addition, there is always an error in the measurement of the distance traveled by the pulse wave, which must be taken into account. Furthermore, children's blood vessels are in a growth phase, and there is some question as to whether this technology can accurately assess the cardiovascular risk of an 11-year-old child.
Therefore, the following assessment items must be added: 1) Echocardiography should be used to assess heart health by observing heartbeat and heart valve motion; 2) Electrocardiograms should be measured to assess problems such as arrhythmias and heart dilation; 3) Exercise stress tests should be performed to assess the effects of exercise on the heart; and 4) The heart should be tested for cardiovascular risk, the effects on the heart during exercise should be assessed.
Comment 4
"Participants whose mothers reported higher fish intake in the first and third trimesters of pregnancy also had significantly higher fish intake in their children." Isn't this a result of the mother's preference being related to the child's eating habits?
Comment 5.
Cardiovascular risk tracking studies should be conducted and evaluated through regular health checks, exercise tests, blood tests, and lifestyle monitoring from childhood through adolescence.

Comment 6.
In the last section of the discussion, the authors state that "The beneficial cardiovascular effects of fish consumption during pregnancy on offspring are still inconclusive." The discussion is disjointed, and I would like to see a little more organization of the preventive effects of maternal intake during pregnancy, the effects of intake after birth, and the effects of intake after adulthood to clarify what they are trying to say from what this study has revealed.
Reviewer 2 Report
Comments and Suggestions for Authors
In this manuscript the authors explore the possible association of maternal fish intake during the 1st and 3rd trimesters of pregnancy was associated to 11-year-old cardiovascular health. The authors observe no significant associations whit maternal fish intake and cardiovascular health in11-year-old. The focus of work is very interesting, and the manuscript is well organized and well written, there are some improvements to be made.
1) The authors collect the diet of the children at 11 years of age, used a true/false 16-item questionnaire based on the consumption of fruits, vegetables, legumes, seafood, cereals, nuts, dairy, and olive oil was. It would be interesting to highlight the results obtained and discuss them.
2) Did the authors analyze the data by separating the children by sex? It would be interesting to report results divided by sex.
Round 2
Reviewer 1 Report
Comments and Suggestions for Authors
The authors responded in good faith to the issues and comments I raised, and improvements were implemented. Consequently, I have given the conclusion that this paper merits publication.